

# Assembling microbial communities: a genomic analysis of a natural experiment in neotropical bamboo internodes

Sonia Ahluwalia[1,2], Iris Holmes[1,3], Rudolf von May[1,4], Daniel L. Rabosky[1] and Alison R. Davis Rabosky[1]

[1] Department of Ecology and Evolutionary Biology & Museum of Zoology, University of Michigan – Ann Arbor, Ann Arbor, Michigan, United States
[2] Feinberg School of Medicine, Northwestern University, Chicago, Illinois, United States
[3] Cornell Institute of Host Microbe Interactions and Disease and Department of Microbiology, Cornell University, Ithaca, New York, United States
[4] Biology Program, California State University, Channel Islands, Camarillo, California, USA

Corresponding author
Iris Holmes, iholmes@umich.edu

## ABSTRACT

Microbes participate in ecological communities, much like multicellular organisms. However, microbial communities lack the centuries of observation and theory describing and predicting ecological processes available for multicellular organisms. Here, we examine early bacterial community assembly in the water-filled internodes of Amazonian bamboos from the genus *Guadua*. Bamboo stands form distinct habitat patches within the lowland Amazonian rainforest and provide habitat for a suite of vertebrate and invertebrate species. *Guadua* bamboos develop sealed, water-filled internodes as they grow. Internodes are presumed sterile or near sterile while closed, but most are eventually opened to the environment by animals, after which they are colonized by microbes. We find that microbial community diversity increases sharply over the first few days of environmental exposure, and taxonomic identity of the microbes changes through this time period as is predicted for early community assembly in macroscopic communities. Microbial community taxonomic turnover is consistent at the bacteria phylum level, but at the level of Operational Taxonomic Units (OTUs), internode communities become increasingly differentiated through time. We argue that these tropical bamboos form an ideal study system for microbial community ecology due to their near-sterile condition prior to opening, relatively consistent environment after opening, and functionally limitless possibilities for replicates. Given the possible importance of opened internode habitats as locations of transmission for both pathogenic and beneficial microbes among animals, understanding the microbial dynamics of the internode habitat is a key conservation concern for the insect and amphibian species that use this microhabitat.

# INTRODUCTION

Host-associated microbial communities are an integral part of terrestrial and aquatic ecosystems, and are essential for the health of many animal and plant species, particularly

with respect to nutrient acquisition and exclusion of pathogens and parasites (*Fitzpatrick et al., 2020*; *Trivedi et al., 2020*). Microbiomes comprise a biological community of bacteria and other single-celled organisms which interact ecologically with each other and their host (*Compant et al., 2021*). Host-associated microbial communities assemble through the process of lineages settling in a location and either establishing themselves or failing to do so. Over time, the survivors of abiotic filtering processes and subsequent interactions between the host and microbial lineages determine the assembly of the microbial community (*Trivedi et al., 2020*). The assembly processes of these communities can be affected by the ability of some microbial lineages to establish early (*e.g.*, priority or incumbency effects) and by the ecological interactions among taxa (*Bittleston et al., 2020*). This dependence on initial conditions, combined with the rarity of truly sterile environments in nature, means that the deterministic processes behind bacteria community assembly can be difficult to study in nature.

Classical colonization theory seeks to explain the processes by which newly accessible habitats acquire a full complement of species (*Horn, 1974*; *Milner, 1987*; *McCook, 1994*; *Milner et al., 2008*). Our current understanding of the development and structure of novel communities is informed by the step-wise processes observed in macroscopic organisms. First, a disturbance occurs which makes new habitats or niches available. These newly empty habitats are often characterized by their sparse distribution of resources, the presence of predators, and/or constraints in the physical environment (*Shea, 2002*). A given species' physiological and ecological characteristics (its functional traits) determine its ability cope with the challenges of the empty habitats and therefore to be an effective colonizer (*Bazzaz, 1979*; *Shea, 2002*). Early successional species establish themselves in new habitats and eventually change the environmental composition through their combined metabolic processes (*McCook, 1994*). Primary colonizers are of special note because they are able to survive in nutrient-sparse environments and often exhibit striking tradeoffs between competitive growth rates and strong colonizing traits (*Cadotte et al., 2006*; *Vellend, 2010*). Next, secondary colonizers recruit to the area and find it hospitable due to the manipulation of resources and nutrients by the primary colonizers (*Vellend, 2010*). Secondary colonizers tend to be stronger competitors than primary colonizers and will eventually replace the primary colonizers as the most abundant organisms in the habitat. Each successive wave of species has higher growth rates and will outcompete the previous successional stage and dominate the new niche (*McCook, 1994*). There is a vast body of evidence to support this sequence in multicellular organisms, which has demonstrated predictive power for the changes in species distribution, abundance, and dominance through early community assembly at the macroscopic scale (*Vellend, 2010*).

On the other hand, at the microscopic level, efforts to determine colonization and succession patterns yield conflicting results (*Nemergut et al., 2013*). These studies are hindered by the difficulty of drawing boundaries around microbial communities given the prolific, perhaps global, dispersal of microbes (*Nemergut et al., 2007*; *Fierer et al., 2010*). Further, it can be difficult to ascertain which taxa contribute meaningfully to the manipulation of the resources of a new niche given the significant metabolic redundancy of microbial communities (*Nemergut et al., 2007*), and microbes' ability to remain dormant

but viable when their local environment is not beneficial to them (*Rehman et al., 2016*; *Sorensen & Shade, 2020*). Despite these differences from multicellular organisms, microbial communities have been observed to adhere to some predictions from the classical ecological theories of colonization and community assembly. Microbial communities tend to follow both classical abundance curves and species-time relationship observations (*Fierer et al., 2010*; *Nemergut et al., 2013*; *Dini-Andreote et al., 2015*). Most relevant to the process of colonization and succession is the observation that microbial taxa may act as ecosystem engineers in early successional stages and alter the environment for future taxa (*Nemergut et al., 2007*).

Host-associated microbiomes are compelling models to study the process of early microbial succession because each new host provides the opportunity to track the entire process of primary succession without waiting for natural disaster or inducing experimental disturbance. In the microbial world, the process of colonization can take days or weeks, rather than years or decades. Plant hosts are particularly interesting because of the interactions between their various microbiota spheres, such as the root-associated bacteria, the leaf-associated bacteria, and bacteria found within plant tissues (*Compant et al., 2021*). The plant microbiome is defined as any area that houses microbiota above or below the soil, as well as around and inside the roots and seeds (*Trivedi et al., 2020*). Host-associated microbes in plants play key roles in nutrient uptake from the soil, tolerance to environmental stress, hormone production, pathogen protection and more (*Schlaeppi & Bulgarelli, 2015*; *Compant et al., 2021*).

The microbial communities that form in small pools of water (phytotelmata) interact with their plant hosts in a variety of ways, including providing metabolic services. For example, phototrophic bacteria in bromeliad tanks may provide nitrogen to the plant (*Vergne et al., 2021*). Conversely, pitcher plants may actively structure their microbial communities in part by providing digestive enzymes to break down prey. The readily available nutrients that result select for microbial communities that can best utilize them (*Grothjan & Young, 2022*). Across a range of habitats, bacteria from the genera Proteobacteria and Firmicutes are often found in phytotelmata (*Vergne et al., 2021*; *Yourstone et al., 2021*).

We present a study of early microbial succession in the water-filled internodes of the Neotropical bamboos *Guadua weberbaueri*, which form large, nearly-monoculture stands within the matrix of Amazonian lowland forest. These woody bamboos develop water-filled internodes as they mature (Fig. 1A). The internodes are near-sterile when intact, but most are eventually opened by vertebrates or arthropods (*Louton, Gelhaus & Bouchard, 1996*). Similar to the related *Guadua angustifolia*, *G. weberbaueri* is a significant component of the Amazonian flora and ecosystem (*Griscom & Ashton, 2006*). *Guadua* internodes are typically exposed to the external environment by foraging brown capuchin monkeys that bite through the walls of the bamboo culm to access the internode (*Davidson et al., 2006*). Other organisms that regularly open the walls of bamboo culms include Rufous-headed woodpeckers and weevils (*Jacobs & von May, 2012*).

*Guadua* bamboo internode microbiota should be reflective of two main sources. First, there will likely be some endogenous microbial colonization from the bamboo's

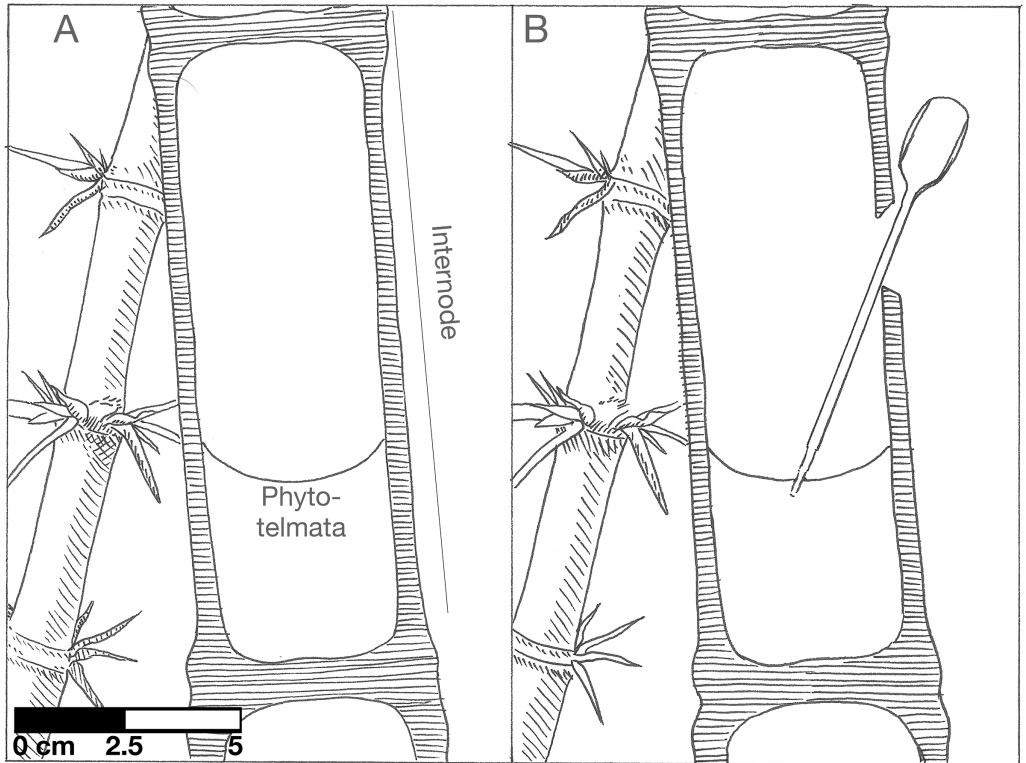

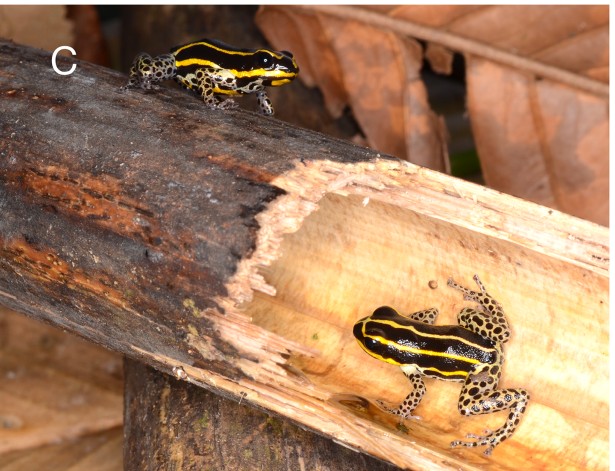

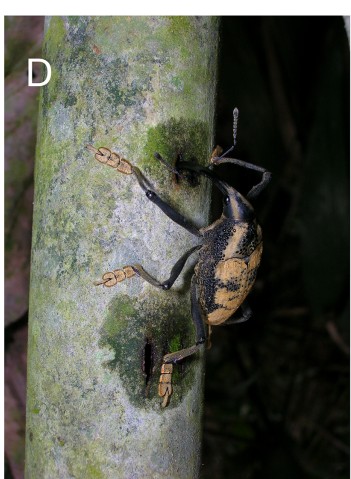

**Figure 1** *Guadua weberbaurei* **characteristics and associated animals.** (A) cross-section of a closed internode partially filled with water (the phytotelmata), (B) a representation of the experimental opening and sampling of bamboo water, (C) the frog *Ranitomeya sirensis*, which lays its eggs in bamboo internodes, (D) weevil feeding on bamboo sap. A and B illustrated by Karin von May, photo credit for C to Rudolf von May, photo credit for D to Marcos Ríos.

rhizosphere. Endospheric microbiota are generally a subset of rhizospheric microbiota (*Turner, James & Poole, 2013*; *Trivedi et al., 2020*), resulting in a non-random bacteria community before initial opening of the internode. After opening, bamboo internodes are often used as key habitats by various vertebrates and invertebrates, such as the poison frog

*Ranitomeya sirensis* (Fig. 1C) and mosquito and damselfly larvae (*Louton, Gelhaus & Bouchard, 1996*; *von May, Reider & Summers, 2009*). Additionally, the scarab beetle *Enema pan* is known to shred and feed on *Guadua* bamboo sap and the shredded portions of the bamboo stem attract many sap-feeding insects (*Jacobs, von May & Ratcliffe, 2012*). These organisms may be potential exogenous sources of microbial colonization along with air and rainfall mediated dispersal and other environmental sources. The small internode pools are therefore a location at which many animal species interact indirectly, and as such could be a transmission point of either pathogens or beneficial bacteria between individuals and species.

We experimentally opened the *Guadua* bamboo internode environment to initiate a bacteria community assembly process (Fig. 1B). We sampled internode water at the time of opening and across a 7-day time series to characterize the early succession of bacteria species colonizing the novel habitat patches. First, we tested whether each internode had a similar set of initial microbial taxa at the time of opening. Second, we tested for taxonomic turnover with time and assessed whether there was an observed loss of initial bacteria OTUs (Operational Taxonomic Units). We hypothesized that there would be taxonomic turnover given previous studies that have documented similar observations (*Nemergut et al., 2013*). In addition, we measured the distance between bacteria communities at each time step to indicate whether stochastic or deterministic processes govern succession in this system (*Dini-Andreote et al., 2015*).

# MATERIALS AND METHODS

## Sample collection

We collected samples of bamboo internode water at Los Amigos Biological Station in southern Peru starting on 30 November 2016 under research permits from the Peruvian National Forest and Wildlife Service (Servicio Nacional Forestal y de Fauna Silvestre) R.D. G. No. 029-2016-SERFOR-DGGSPFFS, R.D.G. 405-2016-SERFOR-DGGSPFFS, and R.D. G. 116-2017-SERFOR-DGGSPFFS. Los Amigos is located in lowland tropical rain forest adjacent to the Madre de Dios river, an Amazon river tributary. We collected water samples from eight bamboo stalks of the species *Guadua weberbaueri* that were distributed across two plots in upland terra firme forest (Fig. 1A). We sampled four stalks from each plot. We determined the following eligibility for internode sampling: the internode must be 1 to 2 m above the ground, contain water, and be closed. We opened the walls of bamboo internodes using a bleach-sterilized blade to cut a 20–40 mm notch horizontally near the top of the node, above the water level. Then we pulled back a section of the internode's external skin using the edge of the blade (Fig. 1B). We chose this method because it mimics the actions of foraging brown capuchin monkeys (*Davidson et al., 2006*; *Jacobs & von May, 2012*).

At the time of opening, we collected 0.5 mL of water from each bamboo internode using sterilized transfer pipettes (Fig. 1B). We preserved these samples in 1.5 mL of CTAB buffer. This buffer has been used in similar studies of microbial communities in other phytotelmata (*e.g.*, pitcher plants; (*Bittleston et al., 2016*)). Our subsequent sampling occurred 24 h after opening, 48 h after opening, and 168 h after opening.

## Nucleic acid extraction

We extracted the total metagenomic DNA from the samples by incubating 200 microliters of CTAB and water mixture with 400 microliters of buffer ATL from Qiagen's DNEasy Blood and Tissue kit and 10 microliters of proteinase K. We incubated the samples at 56 °C for 24 h. We then performed a phenol-chloroform-isoamyl alcohol extraction following *Barker (1998)*. We chose the phenol-chloroform approach to minimize contamination by plant secondary metabolites that could inhibit PCR or sequencing reactions. We included a negative control in the extraction and processed and sequenced the control along with our samples. Samples were amplified using primers specifically developed to target the V3-V4 hypervariable region of the bacterial 16S rRNA gene (*Kozich et al., 2013*). We ran 25 PCR cycles with an annealing temperature of 55 °C. Sequencing libraries were created with the NEBNext DNA Ultra prep kit 2 (cat# E7645L) with barcoded dual-index primers.

Purified and eluted DNA amplicons were stored at 4 °C before being sent for sequencing. We standardized all PCR products using a Qubit measurement and pooled equimolar quantities of each separate PCR well for the final run. Each sample, taken from a single bamboo on a sampling day, was individually barcoded prior to DNA sequencing. Libraries were sequenced using the Illumina MiSeq platform. Data from this project can be found on NCBI's Short Read Archive under BioProject PRJNA766623.

## Bioinformatic pipeline

We used the program mothur v.1.48.0 to identify OTUs (*Schloss et al., 2009*). Following the mothur miseq SOP, we made contigs from our sequences, removed fragments that were not within eight base pairs of our 292 base pair target length, and removed sequences with ambiguous base calls or homopolymers over 8 bp long. We removed sequences that did not align within an 80% similarity cutoff to at least one 16S bacteria sequence in the SILVA v.138 database (*Quast et al., 2012*). We identified the unique sequences in the dataset and removed chimeras. We matched each of our sequences to the RPD Classifier database using the mothur classify.seqs and removed any sequence that aligned most closely with a non-bacteria reference. We then clustered sequences at a 0.03 distance threshold. Finally, we found the consensus classification for our OTUs using the RPD Classifier database v18. The RPD Classifier lists some chloroplast sequences under their "Bacteria" kingdom designation, using the phylum-level labels "Cyanobacteria/Chloroplast." Since we were likely to have bamboo DNA in our extractions, we removed these sequences, resulting in the removal of two OTUs.

We used a custom script in R 4.1.1 to parse the output 'mcc.shared' sample by OTU matrix and 'mcc.0.03.cons.taxonomy' file into an OTU-by-sample matrix in which each cell of the matrix represented the number of reads of an OTU in a specific bamboo-day sample (*R Core Team, 2020*). Both input files and the code used for processing are available Dryad repository (doi: 10.5061/dryad.4qrfj6qd1). To account for possible contamination, we removed any OTU represented by more than 10 sequences in our negative control. Finally, we set all OTUs with a read depth of 10 or less to zero to account for index hopping during sequencing. To account for possible amplification failure, we removed all days of

any bamboo stems that had zero OTUs during any day besides the first. After these steps, we retained seven stems, 19-5, 19-7, 19-8, 19-10, 24-3, 24-6, and 24-9.

## Initial colonizers, taxon turnover, and community diversity over time

To visualize taxon turnover through time, we subsetted the OTU-by-sample matrix by internode and sampling day. To assess patterns of species turnover during early succession, we identified the proportional abundance of each bacteria family in each bamboo stem. We found the mean proportional abundances of all bacteria families grouped by day, and selected the two most abundant in each day, for a total of six families. We plotted the mean abundances of the six focal families across each sampling day. To quantify the species-time relationship, we calculated the Shannon diversity index of each stalk at each sampling point using the R package 'vegan' (*Oksanen et al., 2018*). We tested for significant differences in the diversity values over time using t-test between the distribution of diversity values calculated for each sampling day.

## Community differentiation over time

To identify patterns of community similarity in our dataset, we analyzed beta diversity as a function of time and internode. To do so, we created a Bray-Curtis distance matrix using the R package 'vegan.' To visualize the relative differences between our samples, we performed a nonmetric multidimensional scaling (NMDS) ordination on the OTU-by-sample matrix using the 'vegan' 'metaMDS.' For this analysis, we again used the proportional abundance of OTUs. The function calculates pairwise distances between all sets of communities (in this case each community is a sample from a single internode sampled at a particular timepoint), then represents those pairwise relationships in two-dimensional space while maximizing distances between communities. Since this method is impacted by missing data, we removed samples with fewer than 50 OTUs, thereby removing all of our day 1 samples.

We calculated both the location of each community in NMDS space and the weighted 'species score' for each OTU, representing that OTU's average location in NMDS space. Since many OTUs appear in more than one community, the score represents the location of the centroid of that community weighted by the relative abundances of the OTU. A pattern of samples from individual bamboos grouping together in NMDS space would be consistent with a process of community assembly determined by initial conditions. In contrast, a pattern of samples grouping by sampling day would provide evidence of a more predictable pattern of succession similar to those observed in macroscopic communities. Using the NMDS axes, we plotted the locations of each OTU and visually identified the OTUs belonging to the four most common bacteria taxa. We overlaid the OTU plot with points representing the centroids of each sampled community in NMDS space.

We tested for significant community differentiation through time using a PERMANOVA approach, focusing on the relative importance of sampling time and sampling location. Sampling location refers to the individual bamboo stem from which internodes were sequentially sampled through our experimental procedure. Briefly, this is

a non-parametric approach for MANOVA-like analyses that estimates the proportion of variance in a multivariate dataset explained by each of several factors. PERMANOVA uses a pairwise distance matrix between observations that are grouped according to potential explanatory variables. The approach compares distances between centroids (the mean position of all included points on the axes) of the pre-defined explanatory groups to distances between centroids of randomly sampled observations. The p-value derived from the approach represents the proportion of randomly sampled groups that have centroids that are more separated than those of the explanatory groups. The approach is well-suited to zero-inflated community data because it makes no distributional assumptions about the underlying data (*Anderson, 2001*, *2017*; *McArdle & Anderson, 2001*). We implemented the PERMANOVA test using the 'adonis2' function in 'vegan'.

We applied a square root transform to the OTU-by-sample community matrix, then used the Bray-Curtis index to create a pairwise distance matrix between all sampled communities. We performed 999 permutations of the matrix to find the significance values of location (*e.g.*, bamboo stem) and sampling day as potential explanatory variables for community differentiation. Finally, we calculated the pairwise Bray-Curtis dissimilarity between all samples within each timepoint and plotted those values against a time axis. We performed this analysis to determine whether internodes tended to converge toward a stable community structure over time, or whether they tended to have idiosyncratic community development trajectories at the level of individual bacteria OTUs.

## RESULTS

### Early successional stages and taxon turnover

At the time of opening, only one stem had detectable bacteria, with the two most abundant taxa belonging to the Proteobacteria families *Xanthomonadaceae* and *Acetobacteraceae*. Only two stems had detectable bacteria at this stage. By 24 h after opening the internode (day 2), members of the Proteobacteria families *Enterobacteriaceae* and *Burkholderiaceae* being the most abundant. By 48 h after opening, *Moraxellaceae* had replaced *Enterobacteriaceae* as the most abundant taxa. By 7 days after opening, the two most abundant bacteria taxa were *Comamonadaceae* and *Arcobacteraceae* (Fig. 2A). We found that bacteria community Shannon diversity steadily increased over time (Fig. 2B). Shannon diversity differences were statistically significant between day 1 and day 3 ($p = 0.0012$, t = −5.069, df = 7.3814), day 1 and day 7 ($p = 0.0013$, t = −6.144, df = 5.326), day 2 and day 7 ($p = 0.0168$, t = −3.0121, df = 7.997), and day 3 and day 7 ($p = 0.0323$, t = −2.740, df = 6.261).

### Community differentiation over time

Our sample centroids show a steady progression from day 1 to day 7 across our NMDS axes, with samples from each day generally clustering in NMDS space along axis 1 (Fig. 3A).

The PERMANOVA test of sampling day and stem (*e.g.*, individual bamboo stem) as explanatory variables for changes in community composition differentiation found evidence for significant sampling day but not stem (sampling day: $p = 0.001$, F = 3.011,
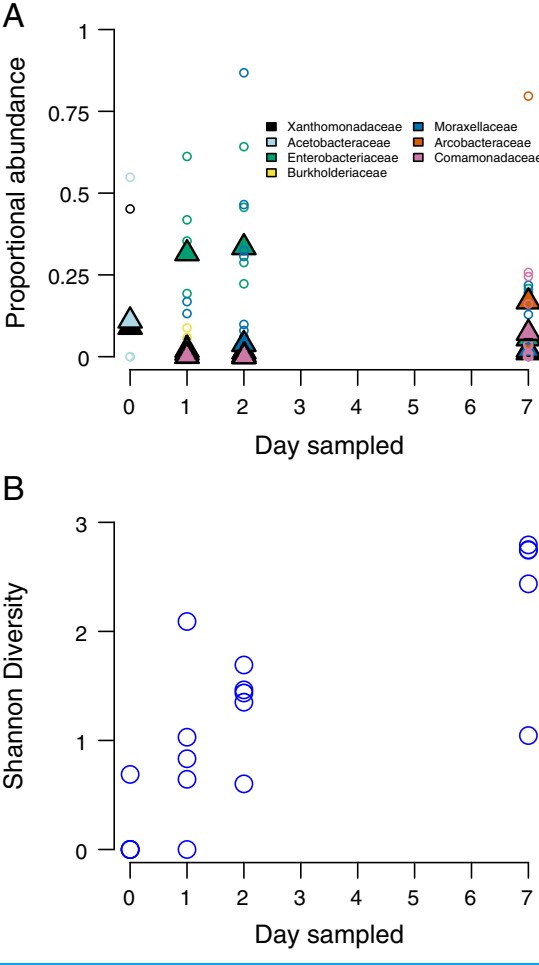

**Figure 2 Species composition and community diversity through time.** (A) abundance of the top four most common bacteria taxa in each sampling day. Triangles represent mean abundances across bamboos, small circles are single bamboo measures. (B) Measures of Shannon diversity of the bacteria community of each bamboo on each sampling day.               

df = 3; stem: $p$ = 0.168, F = 1.180, df = 4). We had four sampling days and seven stems, for a total of 28 samples. Proportionately more variance in community composition was explained by sampling day relative to stem ($R^2$ = 0.415 $vs$ $R^2$ = 0.217). Inspection of Bray-Curtis distances between bacteria communities showed that were high throughout the sampling period, with the exception of our single pairwise comparison at day 0 (Figs. 3B and 3C).

## DISCUSSION

In this study, we were motivated to identify how patterns of early microbial succession compared against theoretical expectations derived from macro-scale organisms and previously studied bacteria community microcosms. Particularly at higher levels of taxonomic resolution, our results reflected classical predictions about early succession derived from macro-scale communities (*Horn, 1974*; *Milner, 1987*; *McCook, 1994*; *Milner et al., 2008*). Our hypotheses regarding common primary colonizers and taxon turnover

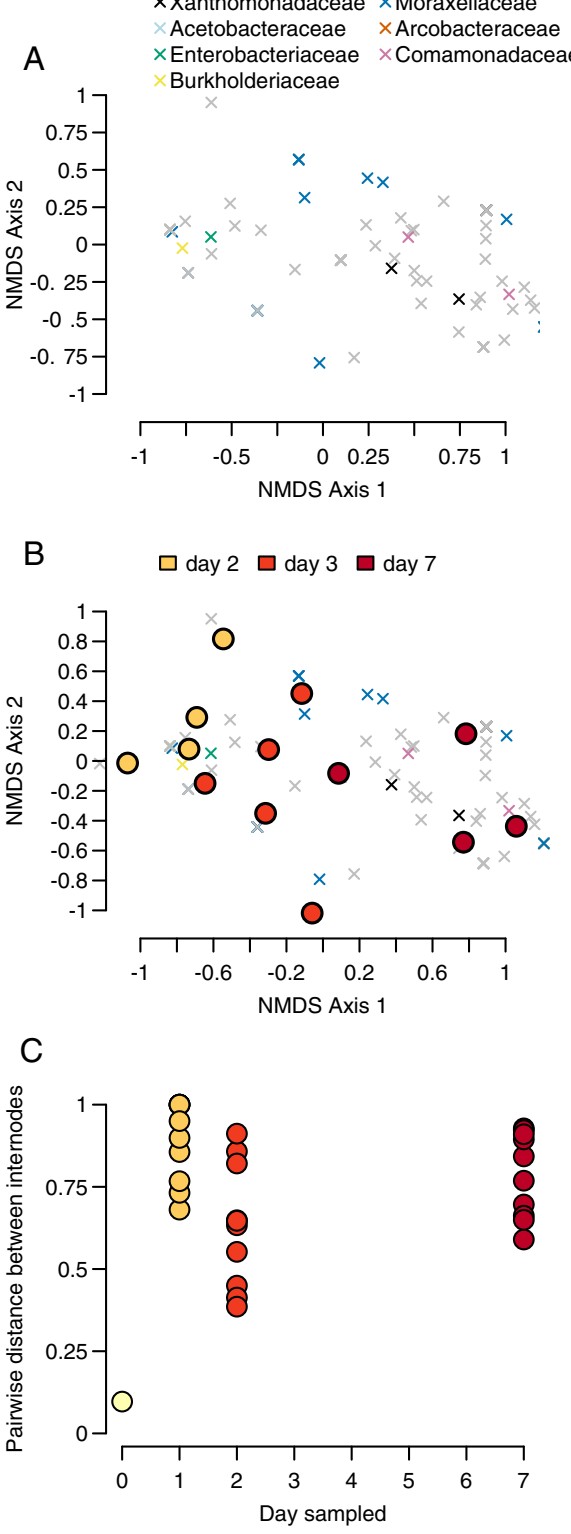

**Figure 3 Community progression through time.** (A) OTUs of the four most abundant bacteria taxa (represented by x-shaped points) cluster in different regions of the NMDS space, (B) centroids of the communities (circular points) for each sample cluster according to day, and (C) pairwise distances between bamboo samples within sampling days become larger over time.

were supported by the consistent primary colonization by *Enterobacteriaceae* and the subsequent domination by other taxa over time (Fig. 2A). We also found that taxon density increased significantly over time, and there was a significant difference in taxonomic diversity between initial and final sampling time (Fig. 2B). This suggests a clear separation in taxonomic diversity and identity between our initial sampling and our final sampling. Recapitulating the turnover analysis performed on OTU counts, we see specific taxonomic groups of bacteria associated with sampling-day clusters along the NMDS axes. Other research has led to similar conclusions that large-scale ecological processes are replicated in phytotelmata microbiomes. For example, tree hole microbial communities show a shift from r- to K-selected species over time (*Pascual-García & Bell, 2020*).

Our PERMANOVA test showed that communities were more differentiated by the day of sampling than by individual bamboo stem, further indicating a predictable successional progression. We found that while the composition of higher taxa in the internodes followed a predictable progression (Figs. 2A and 2B). However, the Bray-Curtis distance between stems within a day was high throughout the sampling period. This result indicates a high degree of stochasticity in community assembly during early succession in bamboo internodes (Fig. 2C). The combination of these results implies a strong cumulative interaction between individual bamboo stem and time that resulted a community structure at the OTU level that was heavily influenced by contingent events in early community assembly despite a much more predictable successional pattern at the bacteria phylum level. This may reflect seeding of later microbial successional stages by insects or other animals using the open internodes, as has been observed in other phytotelmata systems (*Gebuhr et al., 2006*). As a caveat, our work likely recovered only a portion of the total microbial community. First, we did not use a bead-beating approach during our DNA extraction, meaning that DNA from some bacteria with cell wall chemistries that were not vulnerable to our digestion enzyme may not be represented in our sample. Second, we focused on only bacteria rather than viruses, Archaea, or Eukaryotes. Finally, we sampled only the water in the bamboo, while some microbes likely attach to the plant walls rather than occurring in the water column.

Our work has implications for the ecology of the Amazon rain forest in addition to bacteria community succession processes. Woody bamboo species in the genus *Guadua* are a major component of Amazonian flora and ecosystem (*Nelson, 1994*; *Griscom & Ashton, 2006*) and have been present in southwestern Amazonia for at least 45,000 years (*Olivier et al., 2009*). Species in this genus play important ecological and evolutionary roles in the forest community (*Kratter, 1997*; *Griscom & Ashton, 2006*; *Jacobs, von May & Ratcliffe, 2012*), and hold economic importance in the construction of weapons, forestry, construction, artisanship, and cooking (*Jacobs & von May, 2012*). A more complete understanding of this species' associated microbiota may be an important starting point for future studies regarding bamboo conservation and health. Further studies should focus on subsequent successional stages of the microbial community given that open bamboo internodes can hold water for more than 3 months (*von May et al., 2009*).

Future directions of study might include exploring the unique interactions within and around bamboo internodes, such as opening by Brown capuchin monkeys (or

Rufous-headed woodpeckers or weevils) or their use in the life cycles of insects, and how these external factors may contribute to the characteristics of the internal microbiome. A complete assessment of these relationships may include the metagenomic sequencing of bamboo-associated organisms, particularly the external surfaces of associated insects and amphibians. Another important aspect of the internal endosphere microbiome is its potential interactions and sourcing from the rhizosphere (*Trivedi et al., 2020*; *Compant et al., 2021*). Further investigations of this system might include the metagenomic sequencing of the soil and the internal and external roots of the bamboo to better identify sources of primary colonization and give context to the selective pressures within the bamboo internode.

After bamboo internodes have been opened, they become habitat patches in which a variety of multicellular animals interact with microbial lineages. While we have focused on the role of animals in seeding microbial lineages into the internode, the fitness of the animals is also impacted by their interaction with microbial communities. Animals are dependent on their microbiomes for protection from pathogens and parasites, and gut microbiomes help their hosts digest food (*Stough et al., 2016*; *Rosshart et al., 2017*). In frogs, the skin microbiome can prevent mortality due to the pathogenic chytrid fungus *Batrachochytrium dendrobatidis* (*Bd*) as many groups of cutaneous bacteria produce antifungal compounds inhibiting the growth of *Bd* (*Harris et al., 2006*; *Brucker et al., 2008*; *Burkart et al., 2017*). Recent studies conducted in southern Peru have identified at least 20 cutaneous bacteria isolates that inhibit *Bd* growth (*Catenazzi et al., 2018*). In arthropods, the microbiome impacts host fitness and life history, notably by interacting with parasitic infections (*Hamdi et al., 2011*; *Hughes et al., 2014*; *Bouchon, Zimmer & Dittmer, 2016*). The bamboo internode microbial community may play a key role in the ecology of species inhabiting *Guadua* islands in Amazonia as a source for host-associated microbiome lineages. For example, eight species of frogs and a lungless salamander are known to use open bamboo internodes as a retreat site (*von May et al., 2009*; *Whittaker et al., 2015*). Additionally, two of these species, the poison frog *Ranitomeya sirensis* (*von May et al., 2009*; *von May, Reider & Summers, 2009*) and the tree frog *Osteocephalus castaneicola* (*Whittaker et al., 2015*), are known to breed in bamboo internodes with their larvae completing their development in the phytotelmata. Further research can clarify the role internodes play in seeding bacteria into the various hosts that occur in the community.

In addition to its tropical biology applications, our study serves as a proof of concept for using bamboo internodes as a model system for microbial studies. The unique structure of bamboo internodes allows for a natural, but controlled, semi-closed environment that supports ecological interactions between microbes, vertebrates and invertebrates. Bamboo internodes provide a unique set of characteristics as a naturally occurring microcosm for studying bacteria community dynamics. First, the physical structure of the internode changes little after opening, removing a potential set of confounding variables present in other microcosm systems. Because internodes receive little natural sunlight, bamboo internode communities are composed of heterotrophic bacteria, much like soil communities or host-associated microbiome communities (*Fierer et al., 2010*). Unlike soil communities, internodes have defined boundaries (*Fierer et al., 2010*; *Nemergut et al.,*

2013). Unlike host-associated communities, internode communities should be less directly impacted by host-immune driven top-down control (*Jacobs & Braun, 2014*). However, similar to host-associated communities, the animal occupants of the internode provide the majority of the available nutrients to the system and likely the majority of the colonizing bacteria lineages. As such, internode communities provide a natural model system in which bacteria that are similar taxonomically to host-associated communities interact in an environment without host immune regulation. This contrast could serve as an opportunity to investigate the degree to which host immune regulation alters microbial community dynamics. The versatility of the bamboo internode system as a model, as well as the system's evolutionary uniqueness as a high-endemicity series of ephemeral islands in an Amazonian rainforest matrix, make future work on the internode communities an exciting prospect.

## ACKNOWLEDGEMENTS

We thank Leonora Bittleston for providing helpful suggestions for sample collection. We thank the Servicio Nacional Forestal y de Fauna Silvestre, the Amazon Conservation Association, and the staff at Los Amigos Biological Station. We thank Joanna Larson and Greg Schneider for logistical help with the samples. Laboratory work for this article was performed in the University of Michigan Biodiversity Lab. We thank Karin von May for contributing the illustrations *Guadua* bamboo culm and internodes shown in Figure 1 and Marcos Ríos for the photograph of the weevil in Figure 1.

### Funding

This work was supported by a Packard Fellowship to Daniel L. Rabosky, startup funding from the University of Michigan to ARDR, UROP supplementary research funds to Rudolf von May, an NSF GRFP to Iris Holmes, and support from and by the National Institutes of Health and National Institute of Allergy and Infectious Diseases Award T32Il45821 to Iris Holmes. There was no additional funding received for this study. The funders had no role in study design, data collection and analysis, decision to publish, or preparation of the manuscript.

### Grant Disclosures

The following grant information was disclosed by the authors:
Packard Fellowship.
University of Michigan.
UROP Supplementary Research Funds.
NSF GRFP.
National Institutes of Health and National Institute of Allergy and Infectious Diseases: T32Il45821.

## Competing Interests

The authors declare that they have no competing interests.

## Author Contributions

- Sonia Ahluwalia conceived and designed the experiments, analyzed the data, prepared figures and/or tables, authored or reviewed drafts of the article, and approved the final draft.
- Iris Holmes conceived and designed the experiments, performed the experiments, analyzed the data, prepared figures and/or tables, authored or reviewed drafts of the article, and approved the final draft.
- Rudolf von May conceived and designed the experiments, performed the experiments, analyzed the data, prepared figures and/or tables, authored or reviewed drafts of the article, and approved the final draft.
- Daniel L. Rabosky analyzed the data, prepared figures and/or tables, authored or reviewed drafts of the article, and approved the final draft.
- Alison R. Davis Rabosky analyzed the data, prepared figures and/or tables, authored or reviewed drafts of the article, and approved the final draft.

## Field Study Permissions

The following information was supplied relating to field study approvals (*i.e.*, approving body and any reference numbers):

We worked under permits from Servicio Nacional Forestal y de Fauna Silvestre (research permits R.D.G. No. 029-2016-SERFOR-DGGSPFFS, R.D.G. 405-2016-SERFOR-DGGSPFFS, R.D.G. 116-2017-SERFOR-DGGSPFFS). Field experiments were performed at Los Amigos Biological Station, managed by the Amazon Conservation Association.

## Data Availability

The data from this project is available at NCBI's Short Read Archive: PRJNA766623.

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
