# Peer review of "Assembling microbial communities: a genomic analysis of a natural experiment in neotropical bamboo internodes"

_PeerJ, doi:10.7717/peerj.13958_

## Round 0.1 · original submission · Major Revisions

A major revision is required.

Reviewer 1 ·

Basic reporting

In this manuscript, Ahluwalia et al use environmental metabarcoding to follow the establishment of a prokaryotic microbial community in water-filled bamboo semi-sterile internodes. The authors argue that this system offers a good model to study the microbial communities since there are few opportunities to study the establishment of a new micro biome in a natural setting. I think this is a very interesting approach to study a challenging and under-studied concept and I applaud the author’s creativity. I do think the manuscript needs to be re-drafted as there are a number of missing descriptions and key points that should be clarified.


Major Concerns

Materials and Methods:
Lines 164 – 170: Please explain why a DNEasy Blood and Tissue kit was used? There are a number of kits designed specifically for the extraction of DNA from environmental microbes that also remove PCR inhibiting compounds. Why didn’t the authors choose one of those purpose-made kits? Most (all?) of these utilize bead beating to increase the likelihood of microbial cell lysis. Can the authors explain why they chose to forego bead beating? If there is a logical reason then it should be discussed in the manuscript since their methods do not follow the current state-of-the-art.

What were the PCR conditions? What primers were used (include a proper reference)?

Lines 170-172: How many samples or sets were removed? How many were actually used/produced? The answer to the second question appears on line 228 but I think it should appear in the materials and methods since it is a key part of the experimental design.

Line 176: What was used for the negative control?

Lines 177-178: Equimolar quantities of what?

Line 184: The authors need to provide more detail regarding the removal of low-quality sequences. Did they use a program? What parameters were used to separate low- from high-quality?

Lines 192 and 197: How were these OTU’s produced? OTU usually refers to a specific product or process for grouping similar sequences. There isn’t enough detail to determine if the authors truly produced OTUs or if they are using it as a general term for a unique (or nearly unique) sequence that passed their vaguely described quality control process. An alternative methodology which uses nucleotide quality scores to produce unique sequences uses the program DADA2 and the products are called ASV’s. The approaches that produce OTUs and ASVs produce different sets of sequences and researchers usually purposefully choose one over the other. Please clarify.

The authors were very careful in their descriptions (and reasoning) of downstream analyses in the section starting on line 206 (Community Differential Over Time). In the next draft of their manuscript they should be as careful and thorough with their descriptions of molecular and bioinformatics techniques.

Line 199: Where does the analysis with the two most abundant bacterial genera show up in the manuscript? Is this the same as in Figure 3 where 4 genera are mentioned? Please clarify.

Results:
Lines 256-258: What is the evidence that these 16S sequences were all from the bamboo? The way it is written, it appears as though the authors are assuming any OTU from a photosynthetic organism could only be bamboo and dismiss the possibility of a photosynthetic protist or Cyanobacteria because there is no/low light inside the internodes. Please include an analysis of these data.

Figure 1: There should be a scale bar or something to give the reader a sense of the size of these internodes and/or the volume of water found there.

Figure 2A and lines 253-263: What is meant by abundance? Total number of unique OTUs or the occurrence of specific OTUs (i.e. the number of each individual OTU). If it was the latter, these abundances must be expressed as relative abundances. Did the authors do this? It’s not mentioned in the materials and methods. Please rewrite and clarify. Here is a good mini-review on the topic (Gloor et al. 2017, Microbiome Datasets are Compositional: And This Is Not Optional, Front. Microbiol. 8:2224)

There is no mention of Figure 3c in the Results section.


Minor Edits

Line 155: the common name of a species should not be capitalized “brown capuchin monkey”

Lines 192 and 197: The acronym OTU is used on line 192, before it is defined on line 197.

Line 235: define centroid

Experimental design

All comments were included in the "Basic Reporting" section.

Validity of the findings

All comments were included in the "Basic Reporting" section.

Additional comments

All comments were included in the "Basic Reporting" section.

·

Basic reporting

Professional English is used throughout the entire document. Although there are one or two places in the document that a definition added to the text of the manuscript would increase readability (see the general comments section).

The references are balanced. That is to say that the authors incorporate up to date literature in their writings while not excluding older references when relevant.

The article is well written and professional in structure. This includes the main text, figures and general content. Although a link to your data mentioned on line 179 would be useful as searching for BioProject PRJNA766623 yielded no results.

The hypotheses are clearly written, and data presented is relevant to the results.

Experimental design

The research is original and within the aims and scope of the journal. The questions included are well defined and particularly relevant to freshwater microbiology applications. Additionally, the research was performed to an acceptable degree of rigor. Content present within the document is adequately detailed to allow for replication by other researchers with some minor exceptions (see the general comments section).

While it would have been nice to see the research carried out into further timepoints but as the authors have made clear the many potential avenues for future study, I will be watching their work with great interest.

Validity of the findings

Freshwater microbial research is relatively rare, especially for non-crop related plant-based systems. I appropriately view this work as novel. I would have appreciated comparisons of the trends found in bamboo microbial diversity and succession to other similar plant based freshwater systems. Possible examples could include bromeliad tank plants, carnivorous plants, or other freshwater ecosystems.

The statistics used and conclusions stated are linked to the original research and limited to supporting results.

Additional comments

Main Text:
Line 31 – Add a hyphen for consistency to other uses of “near sterile”. In the context of this study what does near-sterile mean? A certain number of OTUs or CFUs expected to be found in a sample? To most people something is either sterile or it is not. A definition of near-sterile in the introduction would be useful for the reader.

Line 37 – What was your reasoning behind choosing OTUs as a measure of taxonomy as opposed to ASVs?

Line 52 - The Abstract Guidance box was minimized and not removed blocking the number line (52), it is visible in the word doc but not it the pdf, remove as appropriate.

Line 102 – The term “microbiota spheres” could use a clarifying sentence as to how they relate to host associated microbiomes.

Line 182 – The mothur program is typically not capitalized unless at the beginning of a sentence. Also add the version number of mothur you used.

Line 186 – Add the release number used to indicate which version of the SILVA database you used.

Line 188 – Indicate which version of R was used. Also was the custom script going to be published as a supplementary file?

Line 256 – Were representative sequences from the Chloroplast taxa run through BLAST to determine if they were bamboo DNA?

Lines 274 through 277 as well as lines 288 to 292 – These are topics would be better suited to the discussion section, not the results section.

Figures:
Figure 1 A-D - A scale bar to indicate plant size would be valuable for the reader. Especially useful when considering the sampling process that mentioned the size of the incision made on line 152.

Overall, this work was done with great care and effort. Time was obviously put into the work, and it shows in the writing. If the above considerations are incorporated into the text of the document, in this reviewer’s opinion the manuscript should be accepted with minor revisions.

---

## Round 0.2 · accepted · Accept

The authors have addressed all the raised points by the reviewers and submitted data in the Dyrad repository for this very interesting study. Link for embargoed data was also shared.